# Evaluating the Color Preferences for Elderly Depression in the United Arab Emirates

Chuloh Jung [1,*], Naglaa Sami Abdelaziz Mahmoud [2], Gamal El Samanoudy [2] and Nahla Al Qassimi [1]

1   Department of Architecture, College of Architecture, Art and Design, Healthy & Sustainable Built Environment Research Center, Ajman University, Ajman 346, United Arab Emirates; n.alqassimi@ajman.ac.ae
2   Department of Interior Design, College of Architecture, Art and Design, Healthy & Sustainable Built Environment Research Center, Ajman University, Ajman 346, United Arab Emirates; n.abdelaziz@ajman.ac.ae (N.S.A.M.); g.elsamanoudy@ajman.ac.ae (G.E.S.)
*   Correspondence: c.jung@ajman.ac.ae

**Abstract:** The elderly are more prone to develop depression from physical, psychological, and economic changes, and 25.7% of the United Arab Emirates' (UAE) elderly population suffer from depression. Color therapy is a widely accepted treatment to solve the depressive symptoms of the elderly. The color preference of the Seniors' Happiness Centre—in Ajman UAE—a residential space for the elderly, could improve the quality of life, including depression symptoms. This paper explored the relationship between the color preference of the resident bedroom space and the depressive symptoms. As a methodology, using color images as stimuli, the physiological and psychological responses of the 86 elderly participants to the proposed color preference of the resident bedroom interiors—observed through a viewing box to simulate 3D space perception—were compared and analyzed to investigate the relationship between the color preference and depression by a survey with the Geriatric Depression Scale (GDS) and Electroencephalogram (EEG) measurement. The results showed that the elderly's preference for warm colors is higher than that of cold colors, and each room needs a different color scheme because the elderly, 65 and above, have different visual characteristics. There was no significant difference between the left and right alpha wave values of the prefrontal cortex of the participant group. The main reason is that the brain waves are minute electrical signals and appear different from person to person. The color scheme on one side of the wall with increased saturation seemed to improve depressive symptoms effectively. It was found that psychologically, healthy elderly reacted positively to the single-color scheme of the Blue cool color, but elderly with depression reacted well to the contrast color scheme of the Blue-Yellow/Red cool color. This study will serve as critical data to propose more color preferences for the Seniors' Happiness Center suitable for the elderly by studying the response to more diverse colors in the UAE.

**Keywords:** color scheme; elderly with depression; color preference; interior design; residential space

## 1. Introduction

According to the United Nations, all the countries in the world are experiencing a dramatic change in the age structure of their population, driven by longer life expectancy and decreased fertility [1]. On a global scale, there were 727 million persons over 65 years in 2020. By 2050, the projected global number of older persons will be over 1.5 billion. This means that those 65 years or over will increase from 9.3% in 2020 to 16.0% by 2050 [2]. In the United Arab Emirates, the percentage of elderly persons, 60 years and above, in 2000 was 5.1% and is expected to reach 23.6% by 2025 [3]. The United Arab Emirates has a population structure in which UAE nationals make up 11.6% while expatriates make up 88.4%, and many expatriates were born in the UAE or lived for many decades [4].

As human become old, their physical functions deteriorate, and they suffer from various age-related chronic diseases and exhibit anxiety and depressive symptoms [5].

Older people are more likely to develop depressive symptoms due to stress from physical, psychological, and economic changes [6]. Even those who previously suffered from depression are more likely to recur [7]. Compared to epidemiological studies in the MENA regions, which demonstrate depression rates ranging from 13% to 18% [8], 25.7% of the UAE elderly population suffer from depression (20.2%) and anxiety (5.6%) [9]. Depressive symptoms are one of the most common mental disorders in old age. Hence, studies should prevent and manage depressive symptoms in old age [10]. Early detection in the United Arab Emirates is difficult because the elderly feel uncomfortable going to the hospital independently and feel reluctant about mental health problems [11]. Negligence of the depressive symptoms will delay recovery and lead to suicide [12]. It has been reported that it is beneficial to combine non-drug therapy with antidepressants to improve depressive symptoms [13]. Different color routines could be partially a factor that affects the body and the psychology of a person, and it might be used to solve psychological characteristics such as depression, but not from direct physical exposure to it [14]. For the mental health of the elderly, the color might make them feel secure and willing to live if it is selected based on their individual preferences [15]. When creating elderly residents, designers should consider that the color preference should have characteristics different from ordinary adults. The elderly prefer colors associated with their cultural backgrounds, while the young select colors with different perspectives, trends, or moods [16]. The color cognition and cultural aspects affect the color selection. In that case, it is possible to create an interior environment that improves the quality of life and recovers depression symptoms [17].

This study aimed to find out the difference in response to the color preference of the residential space concerning the depressive symptoms of the elderly inhabitants. For this purpose, according to the degree of depression in elderly residents 65 years or older, the physiological and psychological responses to the Seniors' Happiness Centre's color preference were compared and analyzed to investigate the relationship between color preference and depression using prepared colored image stimuli, representing their own resident room's picture.

## 2. Materials and Methods

### 2.1. Elderly Depression and Color Therapy

The elderly face various problems that they have not experienced before with aging. In old age, adverse events such as the decline in physical function, loss of social status, anxiety about death, and demise of a spouse affect depressed emotions [18]. In the elderly, hopelessness leads to depressive, negative psychology, including low self-esteem, social atrophy, pessimistic despair about the future, death, and suicidal thoughts [19]. Aged people usually have some symptoms of sadness, and depression is a common mental health-related problem in the elderly. Unfortunately, the level of knowledge held by healthcare professions in its diagnosis and treatment is now significantly below that which would be ideal [20]. Physical aging induces miserable symptoms in the elderly and causes depression, a negative psychological state accompanied by psychological problems, and decreased physical function. Depression in old age makes the daily life of the elderly difficult due to the feeling of helplessness [21]. Unlike other age groups, depressive symptoms in the elderly are more pronounced than atypical symptoms and complaints of depressed emotions, with physical symptoms and cognitive decline. Anxiety and depression individuals are more inclined to choose a hue of gray to describe their feelings. Therefore, it is difficult to diagnose, and it is often mistaken for another disease and receives inappropriate treatment [22].

Recently, active color therapy research was conducted in psychology and medicine. Art therapy has been reported to help people with various illnesses, including depression. Chromotherapy, or colors to heal, was practiced by several ancient cultures, including the Egyptians and the Chinese. Chromotherapy is also known as light therapy or colorology [23]. Therefore, color therapy, derived from the same principle, has been proven to advance depression treatments positively. Color therapy can provide psychological balance

by stimulating the five senses with color to obtain a psychological therapeutic effect and affect metabolism [24]. In particular, colors treat depressive symptoms as effectively in the psychological stability of depressed people. Psychopathological processes are thought to be a significant component in color preferences, in addition to the impacts, as mentioned above, of colors on mood [25]. Depressed humans have a significantly lower retinal response and sensitivity than healthy humans, leading to the same result as removing color vibrations, which is the cause of continuing depressed emotions, which aggravates depression [26]. As such, colors are closely linked to depressive symptoms, so it is essential to control the color of the interior bedroom environment for residents to avoid depressive symptoms [27].

*2.2. Aging and Color Scheme for Residential Space*

As people age, they experience several changes caused by aging. Visual acuity generally weakens after the age of 60, and vision yellowing occurs [28]. Due to vision yellowing, the recognition ability of short-wavelength-series colors is significantly lowered compared to long-wavelength-series colors [29]. Due to the deterioration of the color recognition ability, it is necessary to create an environment that considers the characteristics of the elderly residents, such as giving a difference in brightness or increasing the brightness [30]. Color is an environmental stimulus that humans perceive and stimulate more strongly than form [31]. For the elderly, color can complement the deteriorated sensory function and give a sense of psychological stability when constructing information about the environment [32]. Therefore, the color plan of the elderly resident's interior bedroom must consider the preferred colors and physical changes due to aging. In general, low-saturation colors, often used in Seniors' Happiness Centre in the United Arab Emirates, are a risk factor for the elderly who have a weakened ability to perceive colors [33]. When people become old, their color vision deteriorates. A process known as 'brunescence' occurs when the crystalline lens ages, allowing the hue to become yellowish and saturated.

There is also a study finding stating that the elderly prefer vivid and colorful colors and that as one of the precautions for the color planning of the elderly's interior bedroom, a friendly and soft atmosphere should be created by using warm colors as a whole [34]. Perception and preferences for hue, chroma, and lightness are influenced by age, and color vision disparities caused by inherited color deficits have directly impacted color preferences. Similarly, color vision restrictions caused by lens brunescence should directly impact color choices. Although some researchers have indicated that color selection differs between older and younger individuals as the young follow trends, more recent investigations have discovered that color preference, psychologically, varies between older and younger persons [35]. Therefore, it is necessary to give changes and diversity in consideration of visual characteristics a stable and comfortable atmosphere in the color plan of the Seniors' Happiness Centre for the elderly.

*2.3. The Depression Symptoms and Color Reactions by Electroencephalogram*

In recent years, the investigation of emotional and mental processes and relationships using brainwave characteristics has increased in various fields [36]. An electroencephalogram (EEG) is a way to measure a human's various biological signals and is used as an index to judge psychological reactions and emotions [37]. The measurement of brain waves occurs in all areas of the brain and depend on the frequency band standards: delta waves ($\delta$, 0–4 Hz), theta waves ($\Theta$, 4–8 Hz), alpha waves ($\alpha$, 8–13 Hz), beta waves ($\beta$, 13–30 Hz), and gamma waves ($\gamma$, 30–50 Hz) [38]. Alpha waves are fundamental neurophysiological waves that reflect the brain's stable state (conscious, creative, relaxed, and light meditation). As it is less affected by brain waves in other areas, the alpha wave has been used to measure emotional stability for a long time [39]. An EEG is used to diagnose and treat depressive symptoms, and it is possible to discriminate the depressed state through the resting EEG [40]. It is an objective measurement method for depressive symptoms, not differences due to subjective depressive symptoms or simple mood changes [41]. The brain

waves' simple mood changes—for example, depression—are usually accompanied by the asymmetry of the frontal cortex alpha waves. The activation of the left frontal cortex is associated with pleasant and positive emotions, and the activation of the right frontal cortex is associated with depressive and negative emotions [42]. The left/right activation of the frontal cortex grasp feelings of pleasure, discomfort, and depression. Neuroscience, the difference in EEG asymmetry between the depressive and regular groups, can be used for clinical diagnosis.

Brain waves help analyze and judge reactions and emotions to colors through objective numbers. For instance, in previous studies regarding depressive symptoms, the red color symbolized vitality and had a therapeutic effect [43]. Therefore, psychological therapy effectively treated depressed patients in red interiors and excited patients in blue interiors [44]. The color's uses, in this case, with high brightness, saturation, and warm colors, bring excitement, and low brightness, saturation, and cold colors bring a sense of calm. As such, color and human physiological reactions rely upon psychological reactions, and both reactions and emotions can be measured through brain waves [45].

*2.4. Research Method*

As shown in Figure 1, the conducted literature review and research contribute to understanding the physiological reactions of the elderly to the color of their interior bedroom. In the literature review, previous compared studies on the psychological characteristics of the elderly, their reaction to color, and the color planning of similar centers for the elderly have led to concrete discussion [46]. The authors established the survey research plan and the survey targets based on the literature review results. In addition, they determined the measurement variable and the survey tools for the progress of the research. The selection of the study participants focused on the elderly aged 65 years or older who had no color perception and brain disease problems. The participants recruited were 86 voluntary elderly residents with the cooperation of two Seniors' Happiness Centers, Mushairaf Area and Al Jurf Area, in the Ajman region of the United Arab Emirates. Each of them signed a consent form requesting their names' privacy and refused to take pictures while proceeding in the experiments. The survey participants were classified into two groups, an ordinary range group (group A) and a group showing depressive symptoms (group B) using the Geriatric Depression Scale (GDS) test. A psychological evaluation of the color image of the interior bedroom was conducted through a questionnaire survey using a color image-emotion evaluation tool and an EEG measurement experiment. The survey research was conducted from January to December 2020 while maintaining social distancing, and a total of 80 responses were used for analysis (Table 1).

**Table 1.** Classification of investigation targets.

| Participants | | Level of Depression | | Total |
|---|---|---|---|---|
| | | Group A | Group B | |
| Age Group | 60s | 10 (17.5%) | 3 (10.0%) | 13 (14.9%) |
| | 70s | 34 (59.6%) | 19 (63.3%) | 53 (60.9%) |
| | 80s and Above | 13 (22.8%) | 8 (26.7%) | 21 (24.1%) |
| Gender | Male | 28 (60.9%) | 18 (39.1%) | 46 (52.8%) |
| | Female | 29 (70.75%) | 12 (29.3%) | 41 (47.1%) |
| Total | | 57 (65.5%) | 30 (34.5%) | 87 (100.0%) |

**OBJECTIVE**

Investigate the physiological and psychological responses of the elderly with depression symptoms to the color of the residential space

**LITERATURE REVIEW**

1. Psychological characteristics of the elderly
2. Elderly's reactions to color
3. Existing study on color planning for residential space for the elderly

**METHODOLOGY**

1. Elderly Depression Assessment Tool
2. Single Color Response Measurement
3. Physiological & Psychological Response Measurement

**ANALYSIS**

Survey with 86 elderly residents (65 and above) in 2 Seniors' Happiness Centers
From January to December 2019
Total of 80 responses were used for Analysis.

Analysis Process
1. Geriatric Depression Scale (GDS) test to dive Group A and Group B
2. Psychological Evaluation of the color image of the interior bedroom via Survey
   (Dark-Bright, Cool-Warm, Dull-Clear, Passive-Active, Dislike-Like)
3. Simultaneous EEG measurement experiment

**RESULT & DISCUSSION**
**CONCLUSION**

**Figure 1.** Research process: objective, literature review, methodology, analysis, result, discussion, and conclusion.

*2.5. Investigation Tool*

2.5.1. Elderly Depression Assessment Tool

The degree of depression or simple mood change of the residents was evaluated using Yesavage, Brink, and Rose's Geriatric Depression Scale (GDS) [47]. There were 30 evaluation entries used in this study, and each statement can be answered simply by using 'Yes/No'. The total score ranged from 0 to 30, where the higher the score, the more serious the depressed state. In this study, the collected answers that range between 0 and 14 points were group 'A', and the answers that range between (15–30) points were group 'B'.

2.5.2. Single-Color Response Measurement

A color chart is indispensable to understand and investigate the participants' preferences for level of color (Preferred, Non-Preferred, and Self-Representation Colors). Thus, based on the Munsell system, the authors created a palette of 10 essential colors (Red (R), Yellow (Y), Green (G), Blue (B), Purple (P), Yellow/Red (Y/R), Green/Yellow (G/Y), Blue/Green (B/G), Purple/Blue (P/B), and Red/Purple (R/P)), in addition to three achromatic colors (White (W), Black (B), and Neutral Gray (nG)) to make a 60 cm × 90 cm hardboard, as shown in Figure 2.

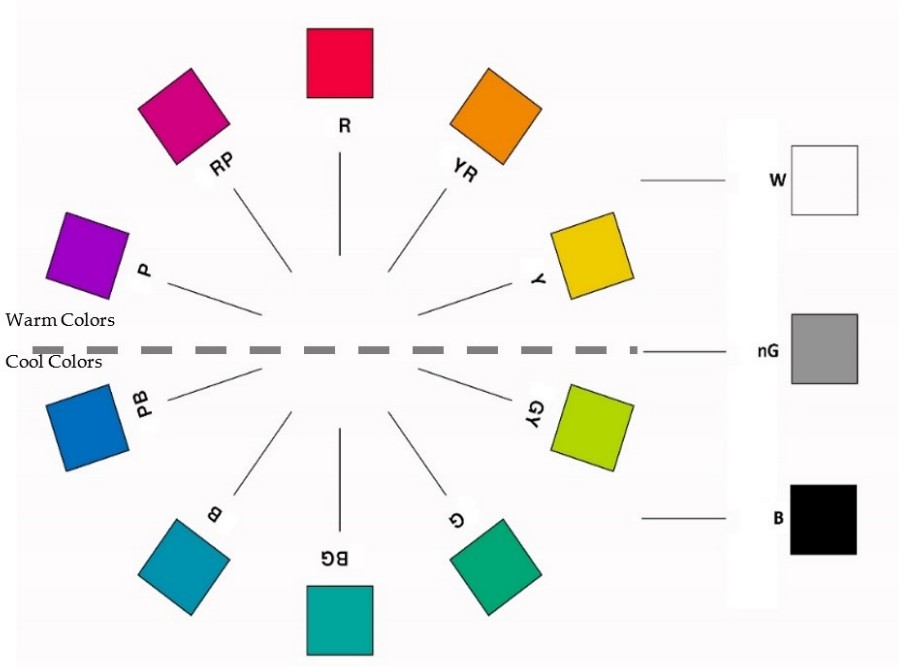

**Figure 2.** Palette of ten essential and three achromatic colors (source: authors, based on the Munsell system-licensed file under the Creative Commons Attribution 3.0 Unported license, 2021).

### 2.5.3. Color Scheme of Resident Interior Bedroom

The suggested color scheme for the resident interior bedroom was the warm colors and their contrast as cool colors, based on the psychological and emotional response standards related to the cultural background [48]. The decisive warm hues selection was R, Y, Y/R, and G/Y, while the cool hues selection was complementary of the warm selections, B/G, P/B, B, and P. The range of choices considered the significance of the warm and cold colors to verify their effects on depression or mood change in previous studies [49]. In addition, the elderly had a lower short-wavelength transmittance due to their visual characteristics [50]. Consequently, using a single-color scheme as a concept aimed to reach psychological stability and contrast colors to create change and diversity [51]. Therefore, the study used a single color (as a front wall: Yellow, Yellow/Red, Green/Yellow, Red, and Blue.

Additionally, the study used a contrast color (as side wall) for the specific single colors, as follows: Yellow-Purple/Blue, Yellow/Red-Blue, Green/Yellow-Purple, Red-Blue/Green, and Blue-Yellow/Red (Figure 3). The achromatic color used was single Grey. The range of brightness and saturation was customized as Vivid (V) tone, which is the range of medium and high saturation, and very pale (Vp) tone, which is the range of high brightness and low saturation (Value 5, Chroma 5), considering the characteristics of interior color and the effect on the EEG. The color application range considered the focal color for the front wall of the interior bedroom, which is visually recognized most, and the secondary color for the sidewall, in the very pale state to blend with the elderly visual conditions (Figure 3). The target residential space was a resident interior bedroom (40 m$^2$) of the Seniors' Happiness Centre, and the samples were produced with the Photoshop program (Color Picker to simulate the Munsell Colors) based on an actual picture from their interior's rooms. It is worth mentioning that the color mode used was CMYK for calibration so that the color values of the selected Munsell were the same on the screen display and when printed in reality.

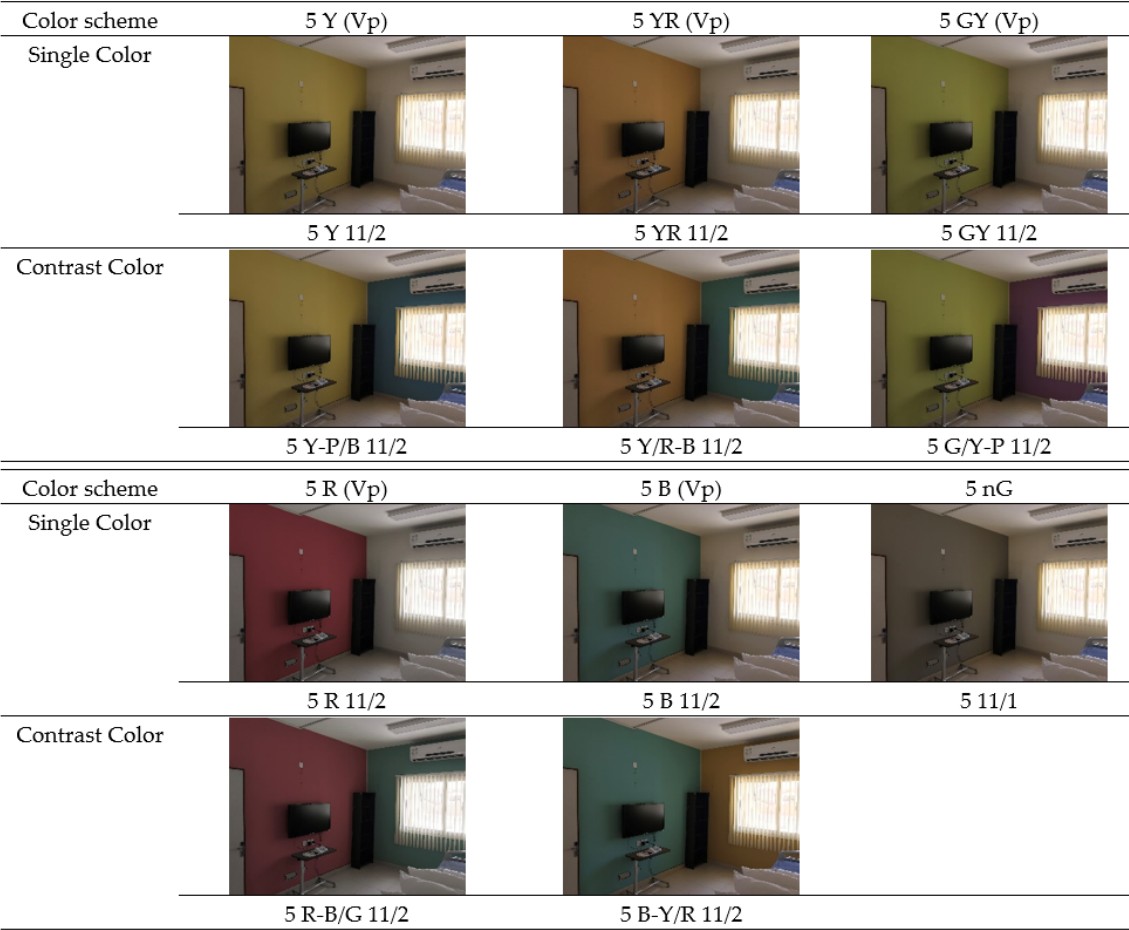

**Figure 3.** The 11 samples of the stimuli of indoor space color scheme each has two versions except Neutral Gray (source: authors, based on an actual elderly room, 2021).

2.5.4. Physiological and Psychological Response Measurement

The physiological response to color was measured using two channels of Neuro harmony S. Two gold-plated dry electrodes were used to measure the EEG in the left and right frontal cortex. Fp1 (left) and Fp2 (right) of the frontal cortex were set as the active electrode, Fpz as the ground electrode, and the ear cortex as the reference electrode [52].

The occipital cortex is the area that primarily acquires visual information in the cerebrum, and the prefrontal cortex is the cortex that understands the acquired color information and is involved in decision-making.

In this study, the EEG value of the prefrontal cortex was used for analysis. The EEG test focused on the experimenter; for accurate data, the first and last stimuli were designated in gray, and stimuli were given for 10 s each. Each color image was stimulated for 20 s, and a gray sheet was displayed for 10 s between color images to reduce the influence of each other (Figure 4). It was calibrated using (Figure 5) Spyder4 Express [53]. According to a particular sequence, the experiment was conducted by maintaining a stable state in the same interior environment where natural light and external noise were blocked. Six pairs of adjectives were placed at the ends of both directions, and a 5-point Likert scale was constructed and evaluated to respond to the participant's thoughts. Three was 'neutral', 2 and 4 were 'slightly', and 1 and 5 were 'extremely'.

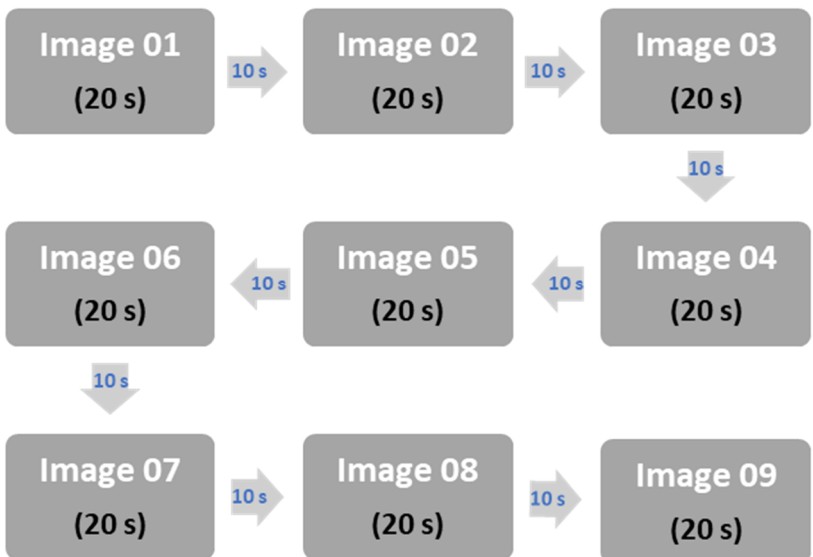

**Figure 4.** The EEG measurement procedure sequence shows the experimental duration.

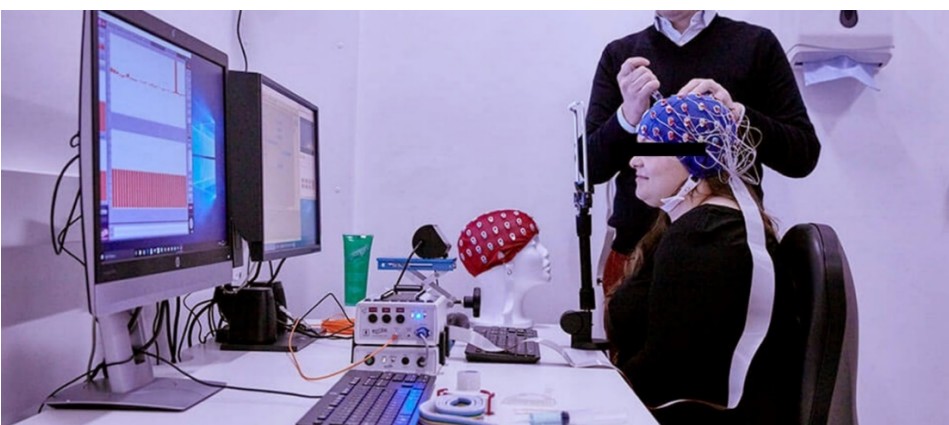

**Figure 5.** The EEG measurement tools, by authors.

## 3. Results

The classification of the 86 participants—where 48 were from Mushairaf Area and 38 were from Aljurf Area—was group A (53 (male = 25, female = 28)) and group B (33 (male = 19, female = 14)). The elderly aged 70 and above were very high, with 63.75% (51 residents), 15% (12 residents) in their 60s, and 21.25% (17 residents) in their 80s or older. The distribution of gender was 52.5% (44 residents) for males and 47.5% (42 residents) for females.

### 3.1. Color Preferencing

The response between residents with depressive symptoms and healthy residents was evident with the single-color preference showing no difference. In this study, the difference between the color preferences of group A and group B was identified, using 10 chromatic basic colors of the Munsell system and 3 achromatic (Figure 1), to *Preferred* and *Non-Preferred* colors, and the *Color which Represents them*.

As a result of analyzing *Preferred* colors in (Table 2) and the statistical analysis of the results in Figure 6, group A showed a somewhat higher preference for cool colors to warm colors in the order of 'Blue' with 13 (26%) and 'Yellow pale' with 7 (14%). The result is consistent with previous studies showing that the preference for colors is related to the country's environment and the cognitive responses of the elderly [54]. For the reason of their preference, participants answered the following for 'Yellow': 'passive' and 'makes the

heart happy'; 'Blue': 'bright color' and 'is always my favorite color'. Nine people (30%) had 'Blue' and four people (13.3%) had 'Yellow' as their preferred color in group B (Figure 6). For the reasons of their preference, they answered the following for 'Blue': 'sea color' and 'it feels good'; 'Yellow': 'the color is deep' and 'it is the sand color'.

**Table 2.** Participants' preferred color, non-preferred color, and representative color.

| Selection/Participants | | R | Y/R | Y | G/Y | G | B/G | B | P/B | P | R/P | W | nG | Black | Total |
|---|---|---|---|---|---|---|---|---|---|---|---|---|---|---|---|
| Preferred Color | Group A | 2 | 5 | 7 | 5 | 4 | 3 | 13 | 4 | 3 | 2 | 2 | 0 | 0 | 50 |
| | | 4 | 10 | 14 | 10 | 8 | 6 | 26 | 8 | 6 | 4 | 4 | 0 | 0 | 100% |
| | Group B | 2 | 3 | 4 | 1 | 3 | 2 | 9 | 3 | 1 | 2 | 0 | 0 | 0 | 30 |
| | | 6.7 | 10 | 13.3 | 3.3 | 10 | 6.7 | 30 | 10 | 3.3 | 6.7 | 0 | 0 | 0 | 100% |
| | All | 4 | 8 | 6 | 6 | 11 | 5 | 20 | 7 | 7 | 4 | 2 | 0 | 0 | 80 |
| | | 5 | 10 | 7.5 | 7.5 | 13.75 | 6.25 | 25 | 8.75 | 8.75 | 5 | 2.5 | 0 | 0 | 100% |
| Non-Preferred Color | Group A | 2 | 2 | 3 | 2 | 2 | 3 | 2 | 2 | 2 | 3 | 2 | 11 | 14 | 50 |
| | | 4 | 4 | 6 | 4 | 4 | 6 | 4 | 4 | 4 | 6 | 4 | 22 | 28 | 100% |
| | Group B | 5 | 0 | 2 | 0 | 0 | 2 | 0 | 1 | 2 | 2 | 2 | 6 | 8 | 30 |
| | | 16.6 | 0 | 6.7 | 0 | 0 | 6.7 | 0 | 3.3 | 6.7 | 6.7 | 6.7 | 20 | 26.6 | 100 |
| | All | 1 | 0 | 5 | 0 | 0 | 3 | 2 | 3 | 6 | 5 | 4 | 17 | 28 | 80 |
| | | 1.25 | 0 | 6.25 | 0 | 0 | 3.75 | 2.5 | 3.75 | 7.5 | 6.25 | 5 | 21.25 | 35 | 100 |
| Color which represent them | Group A | 2 | 2 | 10 | 9 | 6 | 3 | 14 | 1 | 1 | 1 | 1 | 0 | 0 | 50 |
| | | 4 | 4 | 20 | 18 | 12 | 6 | 28 | 2 | 2 | 2 | 2 | 0 | 0 | 100% |
| | Group B | 2 | 0 | 4 | 2 | 3 | 0 | 6 | 4 | 3 | 1 | 2 | 1 | 2 | 30 |
| | | 6.7 | 0 | 13.3 | 6.7 | 10 | 0 | 20 | 13.3 | 10 | 3.3 | 6.7 | 3.3 | 6.7 | 100% |
| | All | 4 | 2 | 14 | 11 | 9 | 3 | 20 | 5 | 4 | 2 | 3 | 1 | 2 | 80 |
| | | 5 | 2.5 | 17.5 | 13.75 | 11.25 | 3.75 | 25 | 6.25 | 5 | 2.5 | 3.75 | 1.25 | 2.5 | 100% |

As a result of the analysis of Non-Preferred colors, the ratio of achromatic colors such as 'White', 'Black', and 'Neutral Grey' was high in both groups. The result is consistent with previous studies that showed that the Non-Preferred for 'Black' is high. Group A had 14 people (28%) in 'Black' and 11 people (22%) in 'Grey'. For Non-Preferred reasons, 'Black' responded with 'despair rises', 'hopeless', and 'dark and gloomy'; and 'Neutral Grey' responded with 'be ambiguous', 'dull', and 'not clear'. Group B had 8 'Black' (26.6%) and 6 'Neutral Grey' (20%). For Non-Preferred reasons, 'Black' had 'dark', and 'Grey' had 'ambiguous' and 'sneaky'.

As a result of the analysis of the colors representing themselves (Table 2) (Figure 6), group A was in the order of 14 'Blue' (28%), 10 'Yellow' (20%), and 9 'Green-Yellow' (18%). For selection reasons, 'Blue' had 'clear and calm', 'clear', and 'unconditionally good'; 'Yellow' had 'bright color' and 'looks good'; 'Green-Yellow' had 'stable color', 'positive', and 'fresh'. Group B had 6 'Blue' (20%), 4 'Yellow' (13.3%), and 4 'Purple-Blue' (13.3%). In the case of 'Blue' as a reason for the selection, they answered, 'to be blue', 'to be seen well', and 'to be bright'. Group A showed a high proportion of cool colors representing themselves compared to the preferred color. Group B showed various preferences and choices for the preferred color and the color representing themselves, regardless of the distinction between gender.

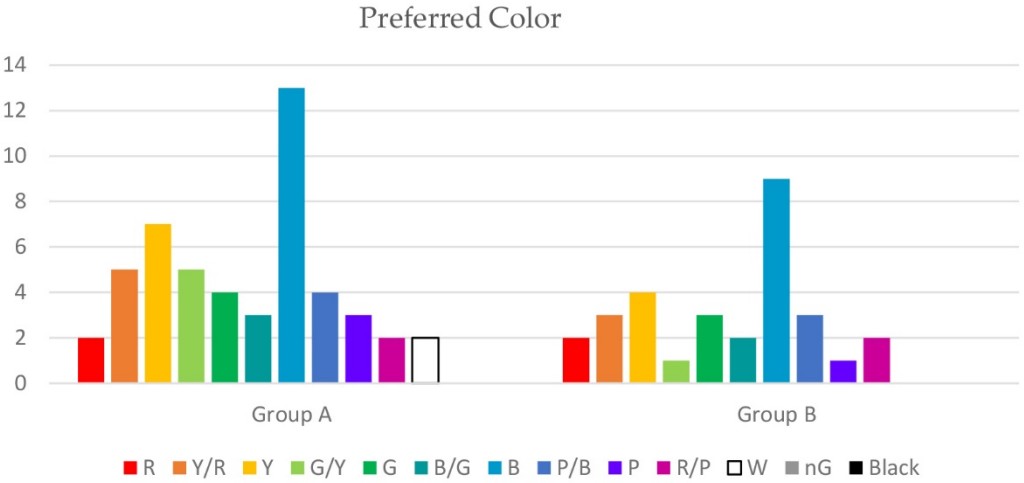

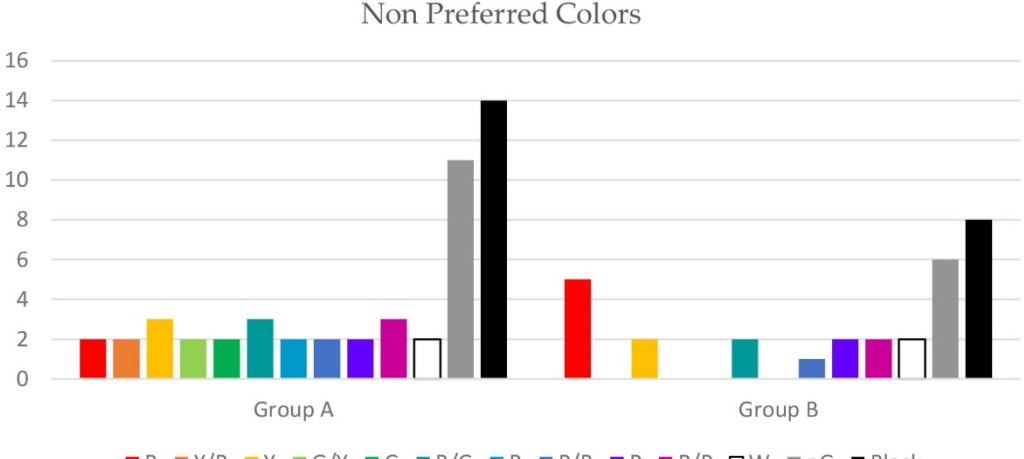

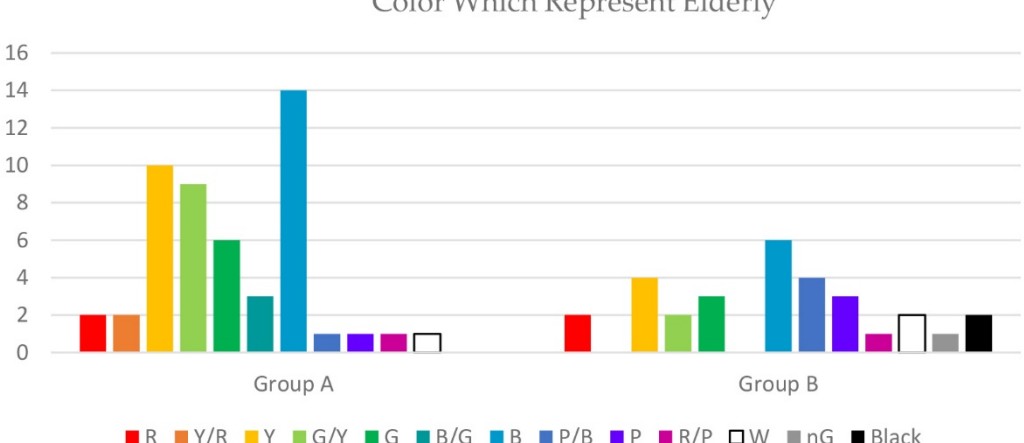

**Figure 6.** Statistical analysis showing the number of 'Preferred', 'Non-Preferred', and the 'Color Which Represents Them' participants, as provided in Table 2.

### 3.2. Psychological Response

Figure 7 shows the evaluation results—using the Likert Scale—of the psychological response to the stimuli of the eleven residents' interior bedroom color scheme images. As a single cool color, the evaluation results for the color Blue were generally similar between group A and group B. For the color Yellow, the average of group A came in the second selection as it was 'bright', 'warm', 'clear', and 'active'. The color Green/Yellow in group A had a lower average of 'clear' than group B. For the color Yellow/Red, the average of group A for 'clear' and 'active' was higher than that of group B. For the color Neutral Grey, the average of group A's 'cloudy' and 'static' was somewhat higher. The color Red, in Group A, had 'dark' and 'dull', while the color Yellow/Red had 'warm' and 'dull'.

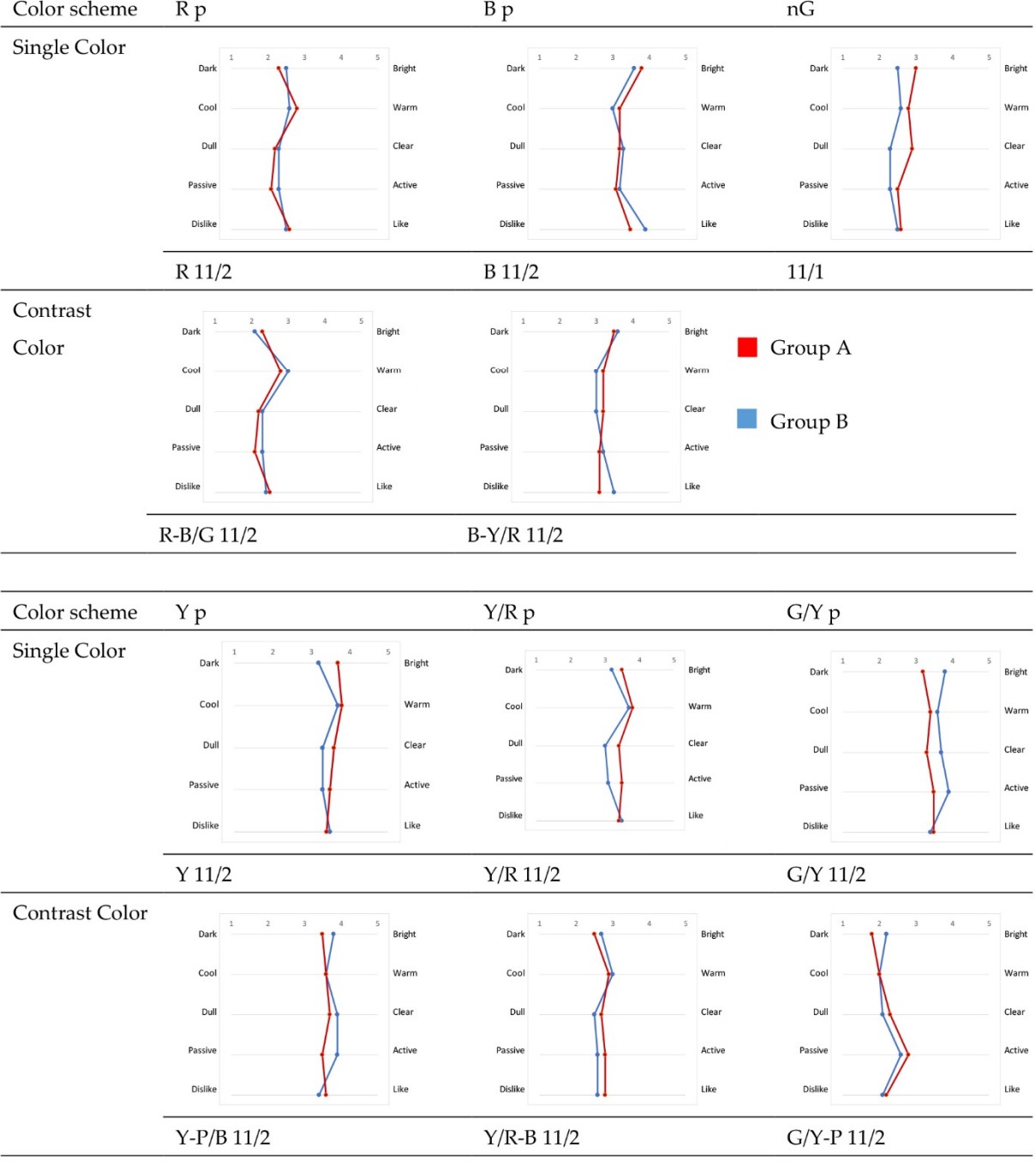

**Figure 7.** Psychological response to interior space colors based on 1–5 Likert scale.

For the contrasting color scheme, the averages of group A and group B for all emotional adjectives were generally similar, and it was found that the average of 'static' in the group was high. In the case of Yellow/Red-Blue and Green/Yellow-Purple, the average of 'dark' and 'cool' was high for cool colors, and the average of 'dark' and 'warm' was high for warm colors. In the contrasting colors, Yellow-Purple/Blue, Red-Blue/Green, and Blue-Yellow/Red, group B's average 'dull' and 'static' were somewhat high.

As for the achromatic color-matching image, the average of 'bright', 'warm', and 'clear' of group B was somewhat higher, and there was no difference between groups for the other 'static-active and 'dislike—like'. The achromatic color-matching image found that, overall, all adjective vocabulary was evaluated close to 'selection 3'.

An independent sample *t*-test was conducted to analyze the difference in the response of 'pleasure' to the color-matching image concerning depression symptoms (Table 3). There was no significant difference between groups except for the achromatic color-matching image in the response score of 'pleasure' for the 11 color-matching stimuli images. In both groups, the score of the cool-color B (M = 3.39, SD = 1.19) was highest, whereas the warm color R (M = 2.62, SD = 1.23) had the lowest score. Looking at each group, the averages of B (M = 3.50, SD = 1.15) for group A and Y/R (M = 3.31, SD = 1.28) for group B were slightly higher. On the other hand, the averages of achromatic color (M = 2.78, SD = 1.22) in group A and (M = 2.60, SD = 1.23) in group B were slightly lower. However, a graph representing the mean response and SD of 'Pleasure' for stimuli color images resulted in a *t*-test of 0.96% (there was no significant (n.s) difference between the groups ($p < 5\%$)) (Figure 8).

**Table 3.** The score of 'Pleasure' for stimuli color images.

| Color Type | Group A M (SD) | Group B M (SD) | Total M (SD) | *t*-Test Result |
|---|---|---|---|---|
| Yellow | 3.40 (1.20) | 3.21 (1.23) | 3.33 (1.21) | −1.642 (n.s) |
| Yellow/Red | 3.38 (1.13) | 3.31 (1.28) | 3.36 (1.18) | −0.947 (n.s) |
| Green/Yellow | 3.34 (1.15) | 3.17 (1.23) | 3.29 (1.21) | −0.454 (n.s) |
| Red | 2.86 (1.21) | 2.55 (1.18) | 2.76 (1.22) | −0.921 (n.s) |
| Blue | 3.50 (1.15) | 3.17 (1.25) | 3.39 (1.19) | −1.068 (n.s) |
| Yellow-Blue | 3.14 (1.20) | 2.52 (1.35) | 2.93 (1.28) | −1.029 (n.s) |
| Yellow/Red-Blue | 2.85 (1.21) | 2.55 (1.18) | 2.76 (1.22) | −0.929 (n.s) |
| Green/Yellow-Purple | 3.00 (1.15) | 3.00 (1.22) | 3.00 (1.17) | −0.973 (n.s) |
| Red-Blue/Green | 2.80 (1.22) | 2.42 (1.17) | 2.62 (1.23) | −0.878 (n.s) |
| Blue-Yellow/Red | 3.14 (1.17) | 2.62 (1.32) | 2.83 (1.20) | −0.929 (n.s) |
| neutralGrey | 2.78 (1.22) | 2.60 (1.23) | 2.82 (1.22) | −0.858 (n.s) |

M: mean, SD: standard deviation.

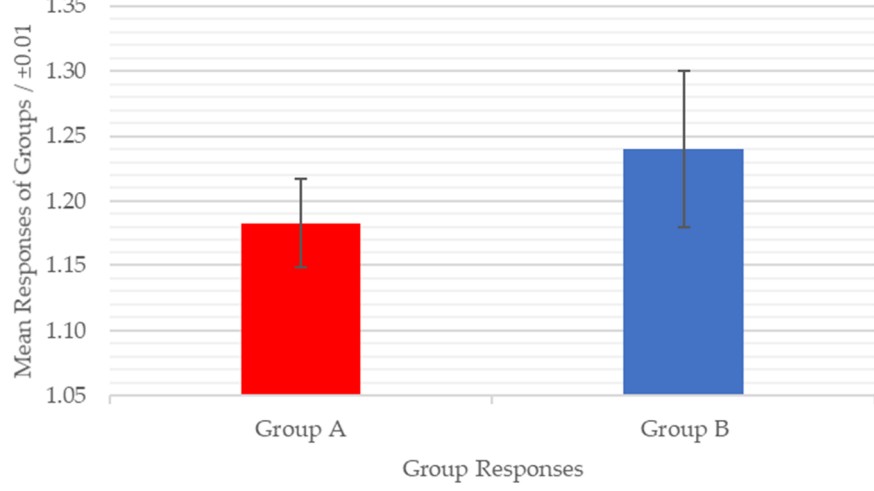

**Figure 8.** Graph of the mean response and standard deviation of 'Pleasure' for stimuli color images.

### 3.3. Physiological Response

An EEG measurement is a precise test, so out of a total of 80 survey residents who participated in psychological characteristics analysis, ten residents in group A (male = 6 people, female = 4 people) and ten residents in group B (male = 4 residents, female = 6 people), a total of twenty residents were selected, and the analysis used the measured results. As a result, the 'male and female' ratio was 50.0%, while the 'age' ratio was 50.0% (10 residents) in their 70s, 35.0% (7 residents) in their 80s and over, and 15.0% (3 residents) in their 60s (Table 4).

**Table 4.** Classification of investigation targets for EEG.

| Participants | | Level of Depression | | Total |
| | | Group A | Group B | |
|---|---|---|---|---|
| Age Group | 60s | 2 (20.0%) | 1 (10.0%) | 3 (15.0%) |
| | 70s | 4 (40.0%) | 6 (60.0%) | 10 (50.0%) |
| | 80s and Above | 4 (40.0%) | 3 (30.0%) | 7 (35.0%) |
| Gender | Male | 6 (60.0%) | 4 (40.0%) | 10 (50.0%) |
| | Female | 4 (40.0%) | 6 (60.0%) | 10 (50.0%) |
| Total | | 10 (100.0%) | 10 (100.0%) | 20 (100.0%) |

According to the color samples, the observation and the measurement showed a change in alpha wave of 8~13 Hz. Alpha wave ($\alpha$) signals of the left (Fp1) and right (Fp2) prefrontal cortexes were collected to analyze the EEG signal for the degree of depression and color-matching images. Alpha waves are brain waves that represent positive and negative emotions in human emotions. As an EEG is a fine electrical signal, it may not be statistically significant, but the difference can effectively identify the emotions. The formula for analyzing the alpha wave asymmetry value (valence value) is as follows: prefrontal alpha wave asymmetry = Fp1(a) − Fp2(a) (Fp1(a): left frontal cortex, Fp2(a): right frontal cortex, a: alpha wave). The Mann–Whitney test compared the alpha wave asymmetry values of group A and group B, a nonparametric test [55]. Table 5 shows the alpha wave asymmetry values for each group's eleven color-matching images.

**Table 5.** The score of 'Pleasure' for color images stimuli.

| Experimental Colors | Group A M (SD) | Group B M (SD) | Total M (SD) | z-Value |
|---|---|---|---|---|
| Closed Eye | 0.66 (0.49) | −0.41 (0.35) | 0.38 (0.96) | −3.782 |
| neutralGrey | 0.63 (1.28) | −0.41 (0.35) | 0.95 (1.93) | −2.268 |
| Yellow | 0.81 (1.33) | −0.20 (1.90) | 1.10 (1.51) | −3.326 |
| Yellow/Red | 0.79 (1.20) | −0.07 (1.14) | 0.38 (0.96) | −3.326 |
| Green/Yellow | 0.70 (1.17) | −0.51 (1.20) | 0.98 (1.80) | −2.297 |
| Red | 0.61 (1.07) | −0.08 (0.54) | 0.82 (1.49) | −3.628 |
| Blue | 0.90 (1.24) | −0.14 (1.39) | 0.60 (0.93) | −3.024 |
| Yellow-Blue | 0.86 (1.32) | −0.53 (1.22) | 0.52 (0.86) | −3.232 |
| Yellow/Red-Blue | 0.84 (1.19) | −0.13 (1.20) | 0.38 (0.96) | −2.797 |
| Green/Yellow-Purple | 0.60 (1.25) | −0.54 (1.53) | 0.31 (1.43) | −3.366 |
| Red-Blue/Green | 0.59 (1.25) | −0.41 (1.28) | 0.76 (1.10) | −2.177 |
| Blue-Yellow/Red | 0.89 (2.10) | −0.05 (0.64) | 0.76 (1.11) | −3.781 |

Valence = Fp1(a) − Fp2(a) (Fp1(a): left frontal cortex, Fp2(a): right frontal cortex, a: alpha wave).

There was a significant difference in the valence's average values for Group A and Group B; Group A had a positive response, and Group B had the majority of negative responses (Table 5). When the eyes were closed, the valence value was 0.66 (SD = 0.49) in group A, and −0.41 (SD = 0.35) in group B. In group A, the valence values of Neutral Gray (M = 0.63), B (M = 0.90), Y/R (M = 0.79), and R-B/G (M = 0.59) were lower than or similar

to the closed eyes values. In addition, the valence values of all other color-matching images were higher than those of the closed eyes. In group B, the values for all color-matching images except G/Y-P (M = −0.54) and achromatic (M = −0.41) were higher than when the eyes were closed.

Group A had the highest valence value for B cool similar color (M = 0.90, SD = 1.24), and the valence value for the similar cold color of B-Y/R (M = 0.89, SD = 2.10) was high. In group B, the valence value of the R-B/G warm contrast color (M = 0.41, SD = 1.28) was highest, and B and Y/R-B (100) were in order. Overall, group A had a high valence value for warm and similar color images, whereas group B had a high valence value for cold and contrast color images, indicating a significant difference in emotional response for each color image. SD was high, representing an increase in alpha brain wave amplitude in the occipital area in the depressed group, which means increasing the functional level and the ability to reduce symptoms of depression.

The analysis target was to detect the positive and negative emotional responses through the asymmetry values of the frontal cortex's left/right alpha wave to the cool color-matching image. The frequencies of single and contrast images showed preference to the single color in both groups. In general, participants of group A experienced positive emotions even when viewing color-matching images compared to group B. The single cool color Blue showed better acceptance where Group A accepted it entire-ly, but in Group B, only eight experienced positive emotions. Unlike cool colors, warm colors showed significant differences. Group A negatively evaluated Red and its con-trast Red-Blue Green with nine and eight participants, respectively. Group B evaluated six and ten negatively for the red color and its contrast, contrary to the Yellow color (Figure 6), respectively.

## 4. Discussion

This study aimed to find the significant differences in psychological and physiological responses to the color image of the resident's interior bedroom according to the degree of preference response in the elderly population. However, there is a limit in proposing as authentic an interior as possible of the Seniors' Happiness Centre color scheme by applying a simple color scheme and expression technique so that the elderly over 65 in the UAE do not have difficulty in evaluating the color scheme of a three-dimensional residential space. In response to the UAE social change, which is gradually becoming older in the future, it proposed reducing and avoiding depression, a severe elderly disease, by preparing an interior bedroom in the Seniors' Happiness Centre color scheme that considers the psychological characteristics of the elderly.

First, as a result of investigating group A and B's preferred colors (Figure 9), it was discovered that group A had a strong preference for cold colors, while group B had a modest choice for bright colors based on their eyesight. The results of group A are not consistent with the results of the previous studies that the elderly's preference, in the far eastern country, for warm colors is higher than that of cool colors—such a result is associated with the cultural color cognition [56]. The UAE elderly spent their lives primarily on the seaside (blue) and the desert (yellow). In addition, the sea represents the expansion of life as it produces their essential food source [57]. The results of both groups are different to the results of the previous studies that college students of different age groups prefer achromatic colors as the degree of depression increases. Such a result can be interpreted as a characteristic of the elderly who prefer bright colors compared to other age groups. In other words, it can be seen that the elderly have different characteristics from ordinary adults in terms of vision, and a differentiated color plan is needed in the elderly space [58]. As a result of examining the non-preferred colors of groups A and B, both groups A and B showed high dislike rates for achromatic colors such as black and neutral grey. It can be seen that both the regular group and the depressed group have a low preference for neutral colors such as grey and negative perceptions of black [59]. Among the achromatic colors, black and neutral grey are highly unfavorable, and black and neutral grey require attention when planning the color of residential spaces. As a result of examining the colors

representing group A and group B, the ratio of cool color was high, and group B showed a low difference according to the color temperature.

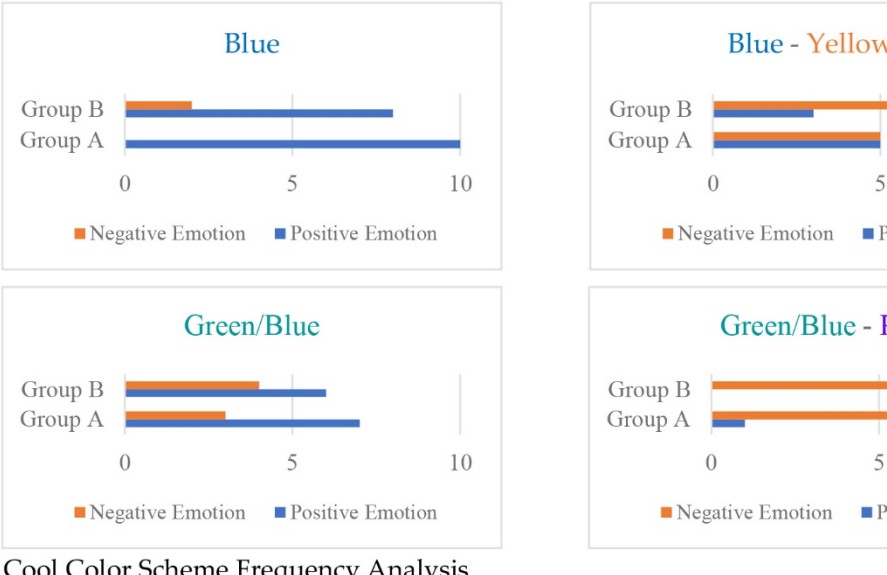

Cool Color Scheme Frequency Analysis

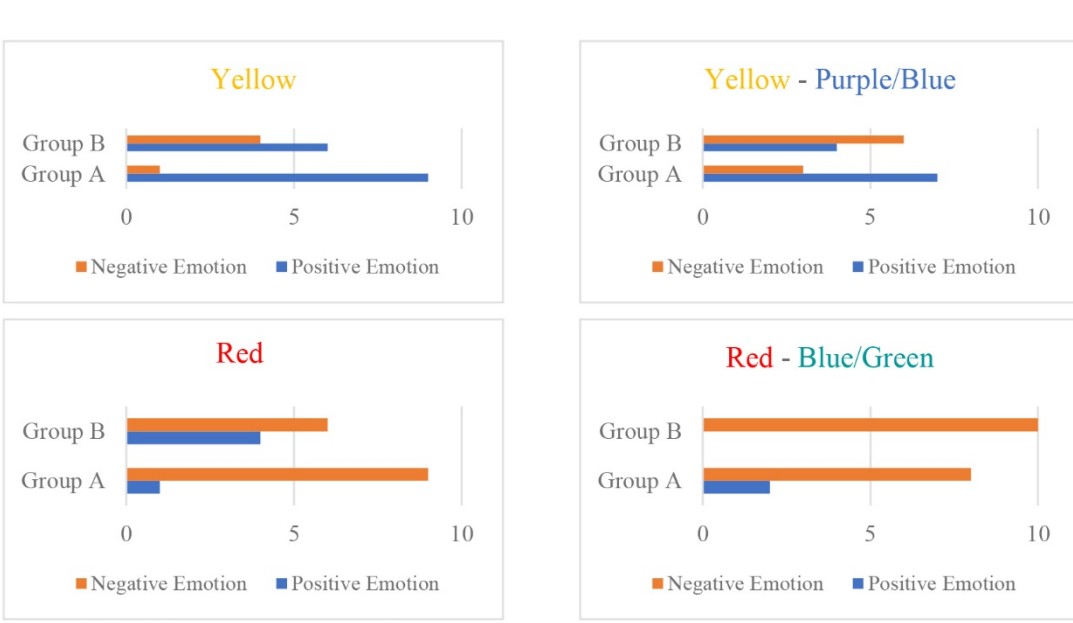

Warm Color Scheme Frequency Analysis

**Figure 9.** Single color vs. contrast color scheme—positive vs. negative emotions frequency's analysis.

Second, it was found that there was little difference in the vocabulary of psychological adjectives for color-matching images according to the degree of depression except for a few items. Such an outcome can be interpreted as having similar psychological reactions to the spatial color image in the case of the elderly regardless of the degree of depression or simple mood change and representing emotions. The mean value of the SD scale for 'pleasure' for the color-matching images of group A and group B differed in the achromatic color-matching image. The single-color score of the cool colors of both groups was high (blue), and the score of the warm color single color was lower (yellow). Looking at each group, the average of the cool color contrast B-Y/R in the B group and the warm color

similar color Y-P/B in the A group was high. Both groups had high psychological reactions to cool colors, so it can be seen that a color plan taking this into account is necessary.

Third, there was no significant difference between the left and right alpha wave values of the prefrontal cortex of group A and group B. This is considered because the brain waves are minute electrical signals and appear different from person to person. However, measuring the valence value for the color-matching image showed significant differences depending on the group. Group A responded positively to the single-color scheme of the B cool color, and group B responded positively to the contrast color of the B-Y/R cool color. Thus, there was no difference in the prefrontal alpha waves of the two groups, but there was a difference in the asymmetry of the alpha waves representing emotions, so it is possible to grasp the subjects' preference for color-matching.

The cultural background and the cognitive responses were found to significantly impact the colors' selection, where single is better than the contrasting [58]. Contrary to the studies performed in Western countries, the cool colors were the preferred colors, especially the sea color, for 26% for standard group A and 30% for depressed group B. The sand color in the warm colors came next in their preferred colors and self-representation by 20% for group A and 13% for group B.

Furthermore, the positive and negative emotions matched the color selection. The correspondence of results proves and accentuates the results of this research.

## 5. Conclusions

This paper investigated the relationship between the color preference of the resident bedroom space, using color images as stimuli, and the depression symptoms using physiological and psychological responses. Understanding the difference in response to the color image of interior resident experimental rooms for color planning of the Seniors' Happiness Centre considering the psychological and physiological reactions of the elderly aged 65 or older in the UAE was a key target. The participants were classified into group A and group B according to the results of the Geriatric Depression Scale and representing the emotional or mood change (GDS) test. Psychological and physiological responses to the single color and twelve color-matching images of the two groups were measured using stimuli images and then analyzed. As for the color-matching image, twelve models were produced from one achromatic color (Neutral Grey), single and contrast colors of warm colors (Yellow, Red, and Yellow/Red), and cool colors (Blue and Green/Yellow). The psychological response to the color image was the average value of the semantic. Furthermore, different method scales composed of adjectives of the color image and the physiological response were measured and analyzed for the prefrontal cortex's EEG alpha wave asymmetry value using a five-point Likert scale.

As mentioned above, psychologically, group A responded positively to the single-color scheme of the Blue cool color, and group B responded positively to the contrast color of the Blue-Yellow/Red cool color. Physiologically, group A responded positively to the single-color scheme of Yellow as a warm color, and group B responded positively to the contrast color scheme of the B-Y/R cool color. Thus, group A showed a common reaction, psychologically and physiologically, favoring a similar color scheme of (Blue) cool colors. However, it can be seen that group B, psychologically, showed a positive response to the contrasting color through their preference for (Yellow/Blue), a warm hue-contrasting color, and they selected the cool hue-contrasting color (Blue-Yellow/Red), physiologically. Following this result, it will be possible to develop an alternative color scheme for residents' interior bedrooms that can be applied in practice. The color scheme on one side of the wall with increased saturation seemed to avoid depressive symptoms or represent emotions effectively.

Future research will propose a more varied color scheme suitable for the characteristics of the elderly by studying the response to more diverse colors. Furthermore, the physiological data of such an investigation need to increase in scope and study to obtain

appropriate and reliable conclusions. Such a section enables future research linking the color physiological effects on the elderly within a greater variety of interior spaces.

**Author Contributions:** Conceptualization, C.J., N.S.A.M., G.E.S. and N.A.Q.; methodology, C.J. and N.S.A.M.; software, N.S.A.M.; validation, C.J., N.S.A.M. and G.E.S.; formal analysis, N.S.A.M.; investigation, G.E.S. and N.A.Q.; resources, C.J. and N.S.A.M.; data curation, C.J. and N.S.A.M.; writing—original draft preparation, C.J.; writing—review and editing, N.S.A.M.; visualization, G.E.S.; supervision, N.A.Q.; project administration, G.E.S. and N.A.Q.; All authors have read and agreed to the published version of the manuscript.

**Funding:** This research received no external funding.

**Institutional Review Board Statement:** The study was conducted according to the guidelines of Ajman University Research Ethics Committee.

**Informed Consent Statement:** Informed consent was obtained from all participants involved in the study.

**Data Availability Statement:** New data were created or analyzed in this study. Data will be shared upon request and consideration of the authors.

**Acknowledgments:** The authors would like to express their gratitude to Ajman University for APC support and Healthy & Sustainable Built Environment Research Center at Ajman University for providing great research environment.

**Conflicts of Interest:** The authors declare no conflict of interest.

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
