# Peer review of "Evaluating the Color Preferences for Elderly Depression in the United Arab Emirates"

_buildings, doi:10.3390/buildings12020234_

Round 1

Reviewer 1 Report

This article is a discussion on the physiological and psychological responses to the stimuli of color images of bedroom interiors of the elderly participants through survey with GDS, preference, semantic differentials, and EEG measurement. Although it has a value in having many numbers of participants in this COVID pandemic with important issue of depression of elderly, I found it is vague, lacking logics and specigicity in some part so needs more work on communication, clarity, logic, expression, and thorough literature review in order to contribute to the body of knowledge.

I have following comments that the author may consider to improve the quality of the paper.

Abstract

Line 19

“The result was that the elderly's preference for warm colors is higher than that of cold colors, and each room needs a different 20 color scheme since the elderly, 65 and above, have different visual characteristics.”

…> I am not sure this can be the result from the study.

48 “Hence, prevention and management of depressive symptoms in old age”

…> grammar doesn’t seem correct.

57 “When creating elderly residents, designers should consider that the color scheme should 57 have characteristics different from ordinary adults”

…>authors may provide how different?

2.1. Elderly Depression and Color Therapy

…>not enough literature review so this chapter seems to need more comprehensive. For example, which color was revealed effective for depression in others studies? In terms of hue, value, and chroma, etc.

-What is weakened ability to “color paper”?

-Please provide any ethical procedure involved in the data collection. How were they recruited? Any compensation given? IRB approved? How long does all experiment take for one participant?

-Discussion can be enhanced. What are the similarities and differences from the results from this research compared to other relevant important literature? Implication to designers, practitioners may be included.

-Provide graphic or photo images of experiment setting will be helpful for understanding, such as photo during experiment setting, photo of participants wearing EEG, image of EEG

Line 195 “the study used Single Color (as a front wall); Yellow, Yellow/Red, Green/Yellow, Yellow, Red, and Blue” …> there are TWO yellows.

-Currently acronyms are very hard to understand or interpret, such as 5Y (Vp), 5R 11/2. What does Vp, 5, or 11/2 mean?

-Single color palette, single color- these are similar and confused. Authors may use different name more easily discernable for better communication.

-Figure 1: need to provide more detailed procedure of experiment, such as how many adjectives were used in psychological response procedure,

-Figure 4: there are total 11 color variations but why nine images (stimuli) used in EEG?

-Line 338 “However, a graph representing mean response and SD of 'Pleasure' for stimuli color images, resulting in a t-Test of 0.96% (There is a significant difference between the group (P <5%) (Figure 7).”

-> which t test was used? Reporting method is not familiar. Does p<5% mean p value is less than 0.05?

In Table 3. what is n.s?

Line 487 “Thus, group A showed a common reaction, psychologically and physiologically, favoring a similar color scheme of cool colors. However, it can be seen that group B showed a positive response to the contrasting color through a preference for warm color-contrasting color psychologically and warm color contrast coloration physiologically.”

  • However the results shows that group A psychologically preferred single B(cool) and phtsiologically singly Y(warm), so above conclusion seems illogical.

Author Response

Dear Revewier 01,

Thank you for your constructive and accurate Criticism.

We did our best to update our manuscript according to your feedback.

Reviewer 2 Report

Comment on Ms #1535850 in Buildings: “Evaluating the Color Scheme for Elderly Depression in United Arab Emirates”

In this paper, the authors investigated elderly participants’ responses to single colours and colour schemes in buildings and were testing the modulation of these effects by depression status. The authors reported both psychological and physiological data, collected with the EEG. Another interesting point is the location where the data are coming from. There is under-representation of non-Western countries in the colour psychology literature. Overall, the article is concisely and clearly written, it is relatively easy to follow the rationale of the study, and merits to be published. Despite being able to recommend it, I would like to raise a couple of concerns.

First of all, the authors, especially in the title, the abstract and the introduction talk a lot about “colour schemes” and their relationship with depression. It becomes almost impossible to know what the authors actually tested until one reaches the method section. It appears that the main measure was colour preferences. Therefore, I suggest stating this very clearly in the title, abstract, and the introduction. It is also important to link the current study with the existing literature in the colour preferences domain. Many relevant previous work is lacking, to name a few, I would say (Hurlbert & Ling, 2007; Jonauskaite et al., 2016; Palmer & Schloss, 2010).

My second point is regarding the strength of evidence for colour therapy. Although the authors give an impression in the abstract and the introduction that it has been shown to have positive effects, the literature is very controversial. This is evident from the cited references. Actually, none of them tested colour therapy per se, they are either theoretical contributions or tests of art therapy. If the authors want to demonstrate a more balanced perspective, then these could be the references to consider (Jonauskaite et al., 2020; O’Connor, 2011). That said, I don’t necessarily think the authors need to make an argument that colour therapy is effective to justify their association and preference study.

My third point is regarding statistical analyses, which are mainly lacking in the article. It is not sufficient to report just percentages, the two groups and the colours should be compared to make any meaningful conclusions. For instance, the results that are discussed in the discussion should be supported with statistics. When statistics are reported, the authors should follow the existing standards on which values to report.

My fourth point is regarding the physiological data. I understand the appeal of these methods. However, I don’t see the added advantage to the current study question. One way would be to justify this approach better. In that case, the sample of 10 participants per group is very tiny and insufficient to draw reliable conclusions. If the authors wish to keep this section, then they should include more data. Alternatively, they could consider eliminating it completely, which would make their story more straightforward.

My fifth point is regarding the discussion. It would be great if the authors could link their findings with the existing literature, just like they did in the introduction section. It is important to understand how their results compare with the context. Also, the cross-cultural aspect is very interesting. The authors briefly mention it in the discussion, but it could be elaborated further by qualitatively comparing the results from the current study and previous studies in Western populations.

Smaller concerns:

  • Abstract: sample size and age should be reported
  • Page 10, line 308: it should read “As shown”
  • Table 2: the digits in the figure should be explained more in addition to the shadings. If this table could also support some statistical analyses, it would be even better.

Dr Domicele Jonauskaite

References:

Hurlbert, A. C., & Ling, Y. (2007). Biological components of sex differences in color preference. Current Biology, 17(16), 3–6. https://doi.org/10.1016/j.cub.2007.06.022

Jonauskaite, D., Mohr, C., Antonietti, J.-P., Spiers, P. M., Althaus, B., Anil, S., & Dael, N. (2016). Most and least preferred colours differ according to object context: New insights from an unrestricted colour range. PLoS ONE, 11(3), e0152194. https://doi.org/10.1371/journal.pone.0152194

Jonauskaite, D., Tremea, I., Bürki, L., Diouf, C. N., & Mohr, C. (2020). To see or not to see: Importance of color perception to color therapy. Color Research & Application, 45(3), 450–464. https://doi.org/10.1002/col.22490

O’Connor, Z. (2011). Colour psychology and colour therapy: Caveat emptor. Color Research and Application, 36(3), 229–234. https://doi.org/10.1002/col.20597

Palmer, S. E., & Schloss, K. B. (2010). An ecological valence theory of human color preference. Proceedings of the National Academy of Sciences, 107(19), 8877–8882. https://doi.org/10.1073/pnas.0906172107

Author Response

Dear Revewier 02,

Thank you for your constructive and accurate Criticism.

We did our best to update our manuscript according to your feedback.

Round 2

Reviewer 1 Report

Overall I think the revision makes the paper clearer and better in communication, but still it needs some improvement in clarity.

In Abstract

  • Although the title is about Color Preferences for Elderly Depression, there is no discussion on the result regarding Color Preferences and Elderly Depression in abstract. Authors may include main finding regarding Color Preferences and Elderly Depression in abstract.

In 2.1. Elderly Depression and Color Therapy

->There could be more discussion on colors revealed effective for depression in others studies.

Line 119

“Although some researchers have indicated that color choice differs between older and younger individuals, more recent investigations have discovered that color preference varies between older and younger persons [35].”

->I am not sure what are the differences between “color choice differs between older and younger individuals” and “color preference varies between older and younger persons”

121

“Therefore, it is necessary to give changes and diversity in consideration of visual characteristics a stable and comfortable atmosphere in the color plan of Seniors' Happiness Centre for the elderly.”

  • Grammar seems not correct

465

“First, as a result of investigating (Figure 9) group, A and B's favorite colors, group A had a high preference for cool colors, and group B showed a low difference in preference according to color temperature.”

  • I am not sure what “a low difference in preference according to color temperature” means.

350

“However, a graph representing mean response and SD of 'Pleasure' for stimuli color images, resulting in a t-Test of 0.96% (There is a no significant (n.s) difference between the groups (P <5%)”

  • How about reporting actual p value?

536

“However, it can be seen that group B showed a positive response to the contrasting color through a preference for (Yellow/Blue) warm color-contrasting color psychologically and warm  color contrast coloration physiologically.”

  • I am not sure what “warm color contrast coloration” means.

Author Response

Dear Reviewer,

Thank you very much for your comments.

We did our best to revise our manuscript based on your feedback.

Thanks,

CJ

Reviewer 2 Report

I thank the authors for making changes to their manuscript based on the reviewer comments. I was rather surprised to get the article back so quickly, which speaks for some of the comments I am making below. I believe the article should be further improved before publication.

Some new sections have been added to the introduction (marked in colour). These sections require some more editing in terms of language errors but also clarity. For instance, one problematic sentence appears on lines 55-57: “The Eelderly prefers the colors associated with their cultural backgrounds, while ordinary (?) adults can (?) select colors with different perceptions, trends, moods, or cognitive background (?)“. The author should take some time to consider if adding these additional blocks of text is actually helpful to their article or have been added just to please reviewer comments.

Most problematic is the section on colour therapy. If the authors have read the articles they are newly citing (O’Connor, 2011; Jonauskaite et al., 2020), they would have noticed that these articles are speaking against the efficacy of the colour therapy. The entire section on lines 51-53 and 85-96 should either be modified to go in line with the scientific evidence or deleted.

The shadings in Table 2 are still not explained. Also, what does “Not Favourite” colour mean? Is this the least favourite or any colour that is not favourite?

The issue with the lack of statistical tests has not been sufficiently accounted for. In particular, the tests should accompany results on lines 257-288 and Table 2.

It is great that the authors are comparing the pleasure variable across the two groups on page 12. The other dependent variables should also be compared statistically. Also, the authors should take into account the issue of running multiple t-tests and control for a family-wise error.

In the discussion, Palmer and Schloss 2010 reference is not appropriate to support the statement in this sentence “Also, the sea represents the expansion of life as it produces their essential food source [57]”

Author Response

(The authors gave the same response as above.)

Round 3

Reviewer 2 Report

Thank you for your changes and good luck with research!

This manuscript is a resubmission of an earlier submission. The following is a list of the peer review reports and author responses from that submission.

Round 1

Reviewer 1 Report

The title of the research is very interesting. However, the methodology in this study can be further improved to enhance the significant outcome for the research question. 

Reviewer 2 Report

A brief summary 

This research investigated the relationship between the colour scheme of the resident bedroom space, using colour images as stimuli, and the depression symptoms using physiological and psychological responses. This paper contributes to the elderly with depression. This paper tried to obtain the responses from the elderly which is a good point in the future application.

General concept comments

In this kind of experiment, preparing stimulus is very important. It must be accurate and precise. It is necessary to describe clearly. The stimuli are colour images. These details should be stated in the last paragraph of the introduction part (lines 60-65) when the aim of the research is presented. It was not mentioned until Section 2.5 (line 187). It was not even mentioned in the abstract. A reader could think that the participants perceived the colour scheme of the resident bedroom, on-site, at the location. 

The presentation of colour image stimuli to the participant should be clarified and given more details. This is important. Perception stimuli as an image or as space can lead to different results. Observing through a viewing box can simulate 3D space perception, that the participants could feel they are making judgments in the location.

The number of stimuli does not cover the main colours. This might lead to a bias conclusion. There are five main hues of Munsell which are Red, Yellow, Green, Blue and Purple and five sub-hue. This research does not balance the selection of hue.

From reading the literature provided in the paper, the control could have been done. For example, the statement "Color therapy can provide psychological balance by stimulating the five senses with color to obtain a psychological therapeutic effect and affect metabolism [24]." , the image stimulus of the achromatic wall could be added to calibrate the EEG signals when participants observe stimuli with and without colour. Neutral grey is achromatic; however, the results show that it is non-favourite compared to white.

Specific comments

  1. The experiment was not described clearly about using Munsell colours for the walls. How do you calibrate that the colour values of the selected Munsell are the same when display on the screen?
  2. How the colours of Munsell were input into Photoshop?
  3. How the display, showing stimulus to participants, was calibrated?
  4. Why only G is skipped? Any reason? 
  5. To my understanding, figure 2 shows only the example of stimulus (as it is written in the figure caption), therefore, in the text, it should be explained about the number of stimuli for all single colour and contrast colour samples. If there are only 11 images, then in the figure caption, it should not be written as "the examples".
  6. The codes in figure 2 such as Y 11/2, what does it mean? It should not be Munsell Value/Munsell Chroma because the colour appearance does not seem so.
  7. Page 5 Line 198 The description is not matched with the figure Yellow/Blue or Yellow/Purple-Blue.
  8. In the text, it was mentioned that Vp represents very pale but in the figure, there is p, not Vp. Are they the same? Also, V represents vivid, however, there is no V found in the figure showing the colour scheme.
  9. Line 199-201. "Vivid tone" which is medium to high saturation should be specified. What is the range of Munsell Chroma of vivid tone? Also, "Very pale" which is high brightness to low saturation, should be described in terms of their Munsell Value and Munsell Chroma.
  10. What is the principle of hue selection for stimuli?